# Droplet Microfluidic Device for Chemoenzymatic Sensing

**DOI:** 10.3390/mi13071146

**Published:** 2022-07-20

**Authors:** Anton S. Yakimov, Ivan A. Denisov, Anton S. Bukatin, Kirill A. Lukyanenko, Kirill I. Belousov, Igor V. Kukhtevich, Elena N. Esimbekova, Anatoly A. Evstrapov, Peter I. Belobrov

**Affiliations:** 1Laboratory of Physical and Chemical Technologies for the Development of Hard-to-Recover Hydrocarbon Reserves, Siberian Federal University, 660041 Krasnoyarsk, Russia; asyakimov@gmail.com; 2Laboratory of Bioluminescent Biotechnologies, Siberian Federal University, 660041 Krasnoyarsk, Russia; k.a.lukyanenko@yandex.ru; 3Laboratory of Renewable Energy Sources, Alferov University, 194021 Saint Petersburg, Russia; antbuk.fiztek@gmail.com (A.S.B.); belousov_k.i@mail.ru (K.I.B.); 4Institute for Analytical Instrumentation RAS, 194021 Saint Petersburg, Russia; an_evs@mail.ru; 5Laboratory for Biomolecular and Medical Technologies, Prof. V.F. Voino-Yasenetsky Krasnoyarsk State Medical University, 660022 Krasnoyarsk, Russia; 6Laboratory for Digital Controlled Drugs and Theranostics, Federal Research Center “Krasnoyarsk Science Center SB RAS”, 660036 Krasnoyarsk, Russia; 7Institute of Silicate Chemistry of RAS, 199034 Saint Petersburg, Russia; ba@inbox.ru; 8Institute of Functional Epigenetics, Helmholtz Zentrum München, 85764 Neuherberg, Germany; 9Institute of Biophysics SB RAS, 660036 Krasnoyarsk, Russia; esimbekova@yandex.ru; 10Department of Biophysics, Siberian Federal University, 660041 Krasnoyarsk, Russia; peter.belobrov@gmail.com

**Keywords:** droplet microfluidics, luciferase, bioluminescence, sensing, chemoenzymatic system

## Abstract

The rapid detection of pollutants in water can be performed with enzymatic probes, the catalytic light-emitting activity of which decreases in the presence of many types of pollutants. Herein, we present a microfluidic system for continuous chemoenzymatic biosensing that generates emulsion droplets containing two enzymes of the bacterial bioluminescent system (luciferase and NAD(P)H:FMN–oxidoreductase) with substrates required for the reaction. The developed chip generates “water-in-oil” emulsion droplets with a volume of 0.1 μL and a frequency of up to 12 drops per minute as well as provides the efficient mixing of reagents in droplets and their distancing. The bioluminescent signal from each individual droplet was measured by a photomultiplier tube with a signal-to-noise ratio of up to 3000/1. The intensity of the luminescence depended on the concentration of the copper sulfate with the limit of its detection of 5 μM. It was shown that bioluminescent enzymatic reactions could be carried out in droplet reactors in dispersed streams. The parameters and limitations required for the bioluminescent reaction to proceed were also studied. Hereby, chemoenzymatic sensing capabilities powered by a droplet microfluidics manipulation technique may serve as the basis for early-warning online water pollution systems.

## 1. Introduction

Nature-inspired multienzyme immobilization by compartmentalization and substrate channeling [1] with digital microfluidic approaches for measuring enzyme kinetics in droplets [2] highlight how hot issues of biological measurements can be solved by the construction of new analytical systems. One of such issues is the control of waste water quality [3,4,5] to prevent reservoirs from pollution by different contaminants such as heavy metal ions, organic molecules, etc. [6,7]. Moreover, performing enzymatic reactions in droplet microreactors is required and very promising to use in different areas of biomedical research [8,9].

Using bioassays for water quality assessment based on bioluminescent bacteria is a well-known concept [10,11,12,13,14]. There are several known disadvantages related to the metabolism of living organisms [15]. The replacement of bacteria with enzymes simplifies the test procedure and substantially increases its reliability [16,17,18].

Chemoenzymatic sensing may be generally defined as the measurement of a change in a biochemical reaction after enzymatic probes and their substrates are placed in the analyzed sample. The group of bioluminescent reactions is convenient for biosensing applications in droplet microfluidics due to the possibility of measuring light signals from individual droplets. The specific chemoenzymatic system which was used in our work consisted of luciferase from bacteria prescribed spelling of species names *Photobacterium leiognathi* as well as NAD(P)H:FMN–oxidoreductase from bacteria spelling of species names *Vibrio fischeri* [19,20] and required chemical substrates. The application of this bioluminescent chemoenzymatic system for ecological monitoring had recently been proposed in a series of studies [21,22,23,24]. The principle of the analysis is to register the quenching of the bioluminescence intensity in the case of pollutants in the samples. A response of the enzymatic probe to pollutants is nonspecific, e.g., as in bacterial [25,26,27,28] or living cell [15] bioassays, and is based on the inhibition of enzymatic activity by toxins. It should be noted that now there are several kits for performing bioluminescent analysis in standard laboratory tubes [24,29,30]. These kits are intended to make single reactions, so it is hard to use them for the continuous monitoring of water quality. Digital microfluidics allow performing biochemical reactions in water-in-oil droplet microreactors continuously generated in fluid streams [31,32]. This can provide the desired automation of measurements to design a continuous water monitoring system. Moreover, such an approach increases the amount of measurements which leads to statistical error reduction. However, the transfer of a biochemical reaction from plastic tubes to water-in-oil droplets raises several issues, such as the influence of the carrying oil phase on the components of a chemoenzymatic reaction, and their diffusion into it [33]. To the best of our knowledge, there have been no studies concerning the influence of the carrying oil phase at the selected bioluminescent reaction.

To encapsulate biochemical reagents, flow-focusing microfluidic droplet generators are usually used. These might consist of a serial connection of microfluidic Y- and X-shaped crosshairs [34]. X-shaped reinjectors moved droplets away from each other in the flow which provides the measurement of each droplet separately [35]. This is important when measuring the optical signals of individual droplets [36]. Similar designs are used in chips for polymerase chain reaction [37,38]. In addition to X-shaped reinjectors, T-shaped reinjectors [39] are used, including high-throughput microfluidic screening platforms [40,41,42,43], as well as in the case of generating femtoliter droplets [44].

The quality of chemoenzymatic sensing with the use of a luminescent reaction depends on the equivalent partitioning of all the reaction components inside the droplets. Mixing in microfluidic devices is a challenging problem due to the absence of turbulent flows. To enhance it, it is necessary to create an asymmetry of flows inside the droplets [45]. One of such mixing methods is to change the speed of the carrier phase at the edges of the droplet [46], for example, by using a tortuous channel [47,48]. Mixing is also possible by deforming droplets on the surface relief of the output microchannel [49].

In this work, our aim was to investigate the performance of the NAD(P)H:FMN–oxidoreductase–luciferase bioluminescent reaction in water-in-oil droplets for developing a continuous water quality monitoring system.

To do this, we developed a droplet microfluidic device for packaging all the components of the system into droplets to measure the bioluminescent signal from each droplet individually. To improve the efficiency of the reaction, we proposed to use the T-junction to enhance the reagents mixing and control the distance between droplets. We then studied how the oil phase affects the reaction, and optimized the phases compositions and chip design to obtain a high level of bioluminescence. Thereafter, we tested our droplet microfluidic device for chemoenzymatic sensing of model pollutants (copper sulfate, 1,3-dihydroxybenzene and 1,4-benzoquinone) and determined their detection limits.

Results were partially presented at the conference “Biosensors 2020” [50].

## 2. Materials and Methods

### 2.1. Reagents

The following reagents were used: lyophilized enzymes of luciferase EC 1.14.14.3 (*Photobacterium leiognathi*) from a recombinant strain of *E. coli* and NAD(P)H:FMN–oxidoreductase EC 1.5.1.29 from *Vibrio fischeri* (Red&Luc) (Laboratory of Nanobiotechnology and Bioluminescence of the Institute of Biophysics SB RAS, Krasnoyarsk, Russia); potassium phosphate buffer (CHEBI: 63036, Fluka, Sweden); reduced nicotinamide adenine dinucleotide (NADH) (CHEBI: 16908, Gerbu, Heidelberg, Germany); ethanol (CHEBI: 16236, Merk, Darmstadt, Germany); tetradecanal (RCHO) (CHEBI: 84067, MP Biomedicals, Solon, France); flavin mononucleotide (FMN) (CHEBI: 17621, Serva, Heidelberg, Germany); mineral oil M5310 (Sigma Aldrich, St. Louis, MO, USA); emulsifier Abil EM180 (Evonik Ind., Essen, Germany); poly(dimethyl siloxane) (PDMS) Sylgard 184 (Dow Europe GmbH, Wiesbaden, Germany); 2075 SU-8 photoresist (MicroChem, Newton, MA, USA); and hydrophobic agent anti-rain repellent (Turtle Wax, Addison, IL, USA).

### 2.2. Preparation of Basic and Starting Solutions

The dispersed phase (droplets content) was mixed in the chip from two original solutions: “basic” and “starting”. The basic solution contained enzymes (Red&Luc) and NADH. The starting solution contained tetradecanal and FMN. It was used to activate the bioluminescent cascade of the enzymatic reactions when mixed with the basic solution.

For the preparation of the basic solution, luciferase 0.05 mg and 0.018 activity units of NAD(P)H:FMN–oxidoreductase from lyophilized preparations were dissolved in 890 μL of 80 mM potassium phosphate buffer with pH 6.8. Thereafter, 4 μmole of NADH was added to the buffer.

For the preparation of the starting solution, 65 μL of tetradecanal 0.25% ethyl alcohol solution was added to 825 μL of water at 25 ${}^{\circ }$C. Then, this was stirred until complete dissolution. After that, 0.5 μmole of FMN was added to the solution.

Two solutions were used as a carrier phase: 1% solution of Abil EM180 in mineral oil; and 0.75% solution of Abil EM180 in 10 mM tetradecanal solution in mineral oil.

### 2.3. Microfluidic Chips

Microfluidic chips were made of PDMS. It was cast into a silicon mold made by photolithography with SU-8 photoresist on a silicon wafer [51]. The channels’ depth was 250 μm. To improve the quality of sealing, PDMS and glass substrates were treated with an air plasma at the atmospheric pressure for 2 min. After that, they were kept at a temperature of 130 ${}^{\circ }$C for 10 min.

Due to the instability of the surface properties of PDMS after plasma treatment and glass hydrophilicity, the inner surfaces of microchannels were coated with a hydrophobic agent anti-rain repellent “Turtle Wax”. It provided a surface wetting angle ≈100${}^{\circ }$. After that, the chips were also treated with mineral oil and kept at a temperature of 130 ${}^{\circ }$C for 10 min to remove the residues of hydrophilic compounds from the pores of PDMS.

### 2.4. Numerical Simulations

Numerical simulations were carried out with COMSOL Multiphysics (Comsol, Sweden) based on the standard model of multiphase fluids (flow and droplet formation) and diffusion of chemical compounds in two dimensions [52]. To calculate the velocity profile of the liquids, the Navier–Stokes equations were solved. Fick’s second law with an added convective term was used to find the concentration distribution.

The two-phase system was described by the level function method [53], where the phase distribution was described by an auxiliary function φ, and the displacement of the boundary between them was determined by solving the mass transfer equation:(1)$$\frac{\partial \varphi }{\partial t}+u\cdot \nabla \varphi =\gamma \nabla \cdot \left(\varepsilon \nabla \varphi -\varphi \left(1-\varphi \right)\frac{\nabla \varphi }{\left|\nabla \varphi \right|}\right),$$
where *u* is a velocity field of fluid (m/s), γ is the reinitialization parameter (m/s), ε is the parameter that controls the thickness of the transition zone (m). The optimal value of the reinitialization parameter is the maximum fluid velocity in the system.

The mixing index, which characterizes the ratio of the dispersion of the reagent distribution, was calculated using the following equation [54]:(2)$$I=\left(1-\sqrt{\frac{\;\int \int \;{\left(c-\bar{c}\right)}^{2}\;dA}{A\cdot \bar{c}\left(c_{max}-\bar{c}\right)}}\right)\cdot 100\%,$$
where the integral is taken inside the droplet; *A* is its surface area (m${}^{2}$); $\bar{c}$ and $c_{max}$ are the average and maximum concentration of the reagent inside the droplet (mol/m${}^{3}$), respectively. When complete mixing is achieved, the index equals 1, and if the mixing is absent, the index equals 0.

In the simulations, a water-like liquid with a density of $\rho_{W}=1000$ kg/m${}^{3}$ and dynamic viscosity $\mu_{W}=0.001$ Pa·s was considered as the dispersed phase. The carrier phase was oil with a density $\rho_{O}=880$ kg/m${}^{3}$, and a viscosity $\mu_{O}=0.03$ Pa·s. The surface tension coefficient at the oil–water interface was σ=0.05 N/m.

### 2.5. Fluid Management

All reagents were sampled into the microfluidic chip at constant pressures in the range of 0–50 kPa using four ITV0010 electro-pneumatic regulators (SMC, Tokyo, Japan) assembled into the microfluidic pressure controller [55]. These regulators were connected to the chip via 15 mL tubes (for basic and starting solutions) and a 50 mL tube (for oil-based medium). The liquids were drawn from the bottom of these tubes using flexible capillaries with an inner diameter of 0.79 mm (Tygon Tubing 1/32 inch). The tubes with reagents were thermally stabilized at 25 ${}^{\circ }$C.

### 2.6. Registration of Luminescence

The output channel of the microfluidic device was connected to a light detector by flexible tubing with an inner diameter of 0.79 mm (Tygon Tubing 1/32 inch). The detector was based on a H7828 photomultiplier tube (PMT) (Hamamatsu Photonics, Hamamatsu, Japan) operated in the photon counting mode. In a light-insulated measuring chamber, droplets moved in a transparent Tygon tubing in front of the PMT aperture. The chamber consisted of two connected plates, between which photographic paper was placed and the joint was filled with black acrylic paint. For the light insulation, the tubing was curved. In front of the PMT window, the photographic paper had a cutout, thus forming an aperture with a width of 5 mm. Additionally, there was a light reflector made of aluminum foil which was placed opposite the aperture under the tube.

The pulses from the PMT were counted with integration period of 0.1 s with the MC74HC393 counter (NXP Semiconductors, Eindhoven, Netherlands) and LPC2103 microcontroller (NXP Semiconductors, Eindhoven, Netherlands) which sent results to an application developed by means of the BlackBox Component Builder framework (https://blackbox.oberon.org (accessed on: 14 July 2022)). The application visualized results for the experiment control and stored the results for further analysis.

### 2.7. Imaging

Imaging the droplet formation in the microfluidic chip was performed with the DM4000 optical microscope (Leica Microsystems, Wetzlar, Germany) using a Manta G223B camera (Allied Vision Technologies GmbH, Stadtroda, Germany). To visualize the flow, microparticles with a nominal diameter of 1 μm (Polysciences Inc., Warrington, PA, USA) or Coomassie Brilliant Blue G-250 dye were added. Images were taken at frequencies of up to 300 fps.

## 3. Results and Discussion

The chemoenzymatic reaction scheme occurred during the chemoenzymatic testing in the water-in-oil droplets generated in the microfluidic device, as shown in Figure 1. Luciferase catalyzes the oxidation of the tetradecanal with the formation of an excited intermediate, which decays to emit light with a maximum intensity at 490 nm. The substrates for this reaction are tetradecanal (RCHO), reduced flavin mononucleotide (FMNH${}_{2}$) and oxygen molecule (O${}_{2}$). FMNH${}_{2}$ is an unstable molecule, and upon contact with O${}_{2}$, it oxidizes to form hydrogen peroxide (H${}_{2}$O${}_{2}$). To maintain FMNH${}_{2}$ at the required concentration, the luciferase reaction is coupled with the reaction of the proton transfer from the reduced nicotinamide adenine dinucleotide (NADH) to the FMN catalyzed by NAD(P)H:FMN–oxidoreductase.

### 3.1. Microfluidic Device for Droplet Generation

Preliminary estimation of light intensity from the bioluminescent reaction showed that the volume of each droplet reactor should be approximately 0.1 μL to fit the PMT sensitivity. We provided several numerical simulations and experimental trials to develop a flow-focusing droplet generator for the formation of water-in-oil droplet reactors with a volume of 0.13 μL (diameter ≈ 0.8 mm) with a generation frequency of 12 droplets per minute (Figure 2a). The tendency to lower the generation frequency was due to the need to increase the residence time of the droplets in the measuring chamber for longer signal collection. This droplet volume is bigger than the one that is commonly used in droplet microfluidics [34,40,44]. Therefore, the improvement of reagents mixing was required to be performed in our work.

#### 3.1.1. Droplet Generation

A previously developed 2D-mathematical model of the droplet generation process in a flow-focusing microfluidic device provided the determination of the droplets’ diameters and their generation frequencies, as well as the velocity field [52]. Confirmation from experimental results (Figure 2b) revealed that the use of such a 2D model allowed us to qualitatively describe the process of droplet formation in the microfluidic chip with shallow channels, the depth of which is less than the width of the droplet formation area.

Numerical simulations raised several important points that should be taken into account during the development of the microfluidic device:Solutions did not mix in the droplets;Changing proportions of pressure of two confluencing liquids of the dispersed phase provide the opportunity to control the composition of droplets;Changing the ratio between the dispersed and carrier phases can possibly generate droplets from 0.5 mm (droplet volume ≈ 0.05 μL) to 1 mm (droplet volume ≈ 0.2 μL). Linear dependence between them was observed as in the case of smaller droplets [52].

Numerical simulation also showed that the formation of droplets with a volume of 0.11 μL occurred at sufficiently close values of the oil and water flow rates. Therefore, the distance between adjacent droplets was far less than their diameter. In order to increase it for independent measurements of the bioluminescent signal from individual droplets, it was necessary to introduce an additional channel for distancing the droplets by adding extra oil between them.

#### 3.1.2. Improving Mixing Efficiency Inside Droplets

To start the cascade of the chemoenzymatic reactions in droplets, it was necessary to mix the basic and starting solutions. The basic solution contained all the components of the chemoenzymatic system, except FMN and tetradecanal, which were in the starting solution to control the moment of the beginning of the reaction. When the flows merged and the droplets were formed, the content of the droplets was not completely mixed. Mixing in microfluidic devices is a challenging problem due to the fact that the fluid flow in microchannels is carried out at Reynolds numbers Re 10${}^{-2}$–10${}^{-3}$ and the flow is laminar. Mixing under such conditions occurs only by diffusion.

It is known that during asymmetric reinjection, the carrier phase is mixed [56]. To study the mixing rate of the reagents, the velocity field of the liquid inside a droplet was obtained while it was passing near the junction with the additional side channel (T-cross) and the additional oil flow. In this region, introduced into the design for increasing droplet spacing, due to the interaction with the lateral oil flow from this side channel, a droplet was significantly deformed. Numerical simulations showed that due to the deformation of the droplet during its passage near the side channel, the formation of asymmetric vortexes (flows) inside the droplet was observed (Figure 3a). These flows led to the intensive mixing of the reagents in droplet, with the mixing index Equation (Equation 2) of up to 60% (Figure 3b) and a high mixing rate of 0.2% per millisecond even after passing through the side channel, when the liquid circulation in the droplet again became symmetric. This is due to the changes in the reagents’ distribution in the entire area of the droplet caused by the asymmetric flows. The results of the experimental studies confirmed the 2D simulation results and are shown in the Figure 3c–e.

The numerical model of the perpendicular inflow of an additional volume of the carrier phase into the flow with droplets without changing the channel clearance showed that opposite the injection site, the droplets were washed by the carrier flow from different sides with different velocities, which led to the significant asymmetry of their Taylor vortices inside the drops and the mixing of their contents. The simulations predicted that the distance between the droplets increased by introducing an additional oil phase from the side channel. By adjusting the pressure, the distance between the droplets could be varied in the range of 0.1–5 mm. Experimental verification confirmed the calculations for the same pressures as in the simulations. Furthermore, during the experiment, a wider range of pressures was tested, at which we provided the spacing of droplets to a distance of 25 mm (see Section 3.1.3 for details).

#### 3.1.3. Final Chip Design

After the experimental studies of droplet formation and mixing efficiency, the final design of the microfluidic device for the encapsulation of the multienzyme system and chemical reagents into water-in-oil droplets was developed (Figure 4). The depth of all channels of the chip was 250 μm. This consisted of liquid connectors, hydraulic resistances, a flow-focusing droplet generator and a droplet content mixer (Figure 5a). The experimental setup allowed to adjust the distance between droplets (Figure 5c). The basic and starting solutions with the components of the multienzyme system were combined together at the Y-cross into a single flow. Furthermore, this flow was dispersed into separate droplets in the droplet generator and all the components of the enzymatic system became encapsulated in these droplets. It was necessary to use an additional channel (T-shaped reinjector) for increasing the distance between the droplets by adding the carrier phase. Moreover, this additional flow deformed the droplets and increased the mixing of the solutions inside the droplets after their generation. Thus, a flow of droplet reactors with a chemoenzymatic system immobilized inside each droplet was formed.

The developed chip was connected to a microfluidic pressure controller through thermostabilized reservoirs with reagents. The inlet channels were equipped with additional hydrodynamic resistances to adapt the chip to the operating pressure range of the microfluidic pressure controller. The pressures were selected so that droplets with a volume of 0.12 μL were formed, the distance between them was 2–2.5 cm and the frequency of generation was 0.2 Hz (Figure 5b). The distance of 2–2.5 cm between droplets was chosen in such a way that only one droplet would be exposed to the PMT detector for measurement through the aperture and there would be no illumination from the neighboring droplets.

### 3.2. Bioluminescent Reaction Specificity in Droplets

A tetradecanal molecule contains a large hydrophobic group and weak hydrophilic group. Thus, it is a weak surfactant that is practically insoluble in water. Due to the chemical transformations of tetradecanal, the droplet contained tetradecanal (C${}_{14}$H${}_{28}$O) with melting point 30 ${}^{\circ }$C, 1-tetradecanol (C${}_{14}$H${}_{30}$O) with melting point 39 ${}^{\circ }$C and tetradecanoic acid (C${}_{14}$H${}_{28}$O${}_{2}$) with melting point 53.8–58 ${}^{\circ }$C. Due to experimental temperatures of approximately 25 ${}^{\circ }$C and the poor solubility of these chemicals in water, they are very unevenly distributed within a droplet. Furthermore, these chemicals have fair solubility in the oil phase; therefore, they might diffuse through the phase interface in both directions (Figure 1). To figure out the diffusion efficiency and dependence of the luminescence intensity on the presence of the tetradecanal in both phases, we added it into the water-based starting solution (W) and into the oil-based carrier phase (O) with the following options:(1)Tetradecanal dissolved in the carrier phase and in the dispersed phase (W+O+). To dissolve tetradecanal in water, it was first dissolved in ethanol and only then an alcoholic solution of the tetradecanal was diluted with water;(2)Tetradecanal dissolved only in the carrier phase (W-O+), while ethanol was added to the dispersed phase without tetradecanal;(3)Tetradecanal only dissolved in the dispersed phase (W+O-).

The concentration of the tetradecanal in the carrier phase was 10 mM, and in the dispersed phase, it was 0.33 mM (taking into account the 1:1 dilution of the starting solution with the basic solution, where the tetradecanal concentration was 0.66 mM).

After the generation of droplet reactors in our device, their luminescence intensities were measured. It was found (Figure 6a) that when tetradecanal was only in the aqueous phase, a light signal was recorded practically at the level of the dark noise of the detector—at approximately 20 RLU. In the case of the tetradecanal dissolution only in the non-polar carrier phase, the luminescence intensity was approximately 340 RLU. The presence of tetradecanal in both phases resulted in a maximum luminescence of approximately 7000 RLU (Figure 6b).

It is known that the luminescence intensity of a multienzyme system first reaches its maximum, after which it slowly declines [57]. Thus, it was necessary to find the position of the detector on the tube. We measured the intensity in 20 s after droplet formation that corresponded a tube length of 25 cm.

Figure 6b shows the signal recorded by the PMT when the tetradecanal was diluted in both phases. Each peak corresponds to an individual droplet passing the detector aperture. The luminescence signal was recorded approximately 16 times per droplet passing the detector.

### 3.3. Chemoenzymatic Sensing for Water Pollution Monitoring

We studied how model pollutants copper sulfate (heavy metal salt), 1,3-dihydroxybenzene (phenol) and 1,4-benzoquinone (quinone) influenced the bioluminescent intensity in droplets. Distilled water GOST R 58144-2018 was used as a control sample. To avoid the cross-contamination of the samples, the microfluidic chip was replaced after the measurement of different pollutants. A total number of three chips were used, one for each type of pollutant.

From Figure 7, it is possible to conclude that the limit of detection (LOD) for CuSO${}_{4}$ was ≈5 μM. The luminescence intensity of 1,3-dihydroxybenzene decreased by a quarter at a concentration of 10 μM and remained at this level even at concentrations over 500 μM, which we attribute to the fact that 1,3-dihydroxybenzene escaped from the droplets into the oil. The LOD for 1,4-benzoquinone was 2 μM.

Our results (Figure 7) show that continuous biotesting in droplet microreactors with the enzymatic system from luminous bacteria can be used to detect some pollutants in water at their MPC level: 1 mg/L for copper sulfate (6.27 μM for molar mass 159.6 g/mol); and an MPC level of 0.2 mg/L for 1,4-benzoquinone (1.85 μM for molar mass 108.1 g/mol). However, the registration of 1,3-dihydroxybenzene encountered certain difficulties. The bioluminescent system used in this work was initially very sensitive to 1,3-dihydroxybenzene, but this sensitivity sharply decrease by more than four orders of magnitude when packed into an emulsion. This could be explained by the high solubility of 1,3-dihydroxybenzene in the oil phase. Less sensitivity for 1,4-benzoquinone could be explained by increasing the reaction yield time at the luminescence maximum. Thus, the reaction might not have had time to reach the maximum luminescence intensity within 20 s after stirring when reaching the photodetector.

## 4. Conclusions

Here, we developed a microfluidic device for the generation of water-in-oil droplets with a bioluminescent bacterial system for continuous chemoenzymatic biosensing applications. The optimization of the reaction parameters provided a reduction in the amount of enzymes and substrates per droplet, preserving the sensitivity of the system to pollutants and the effective mixing of reagents.

One of the consequences of the increase in the volume of droplets was the lack of mixing of their content after the formation of droplets. Carrier phase asymmetric reinjection was used to improve mixing inside the droplets. It was shown that during asymmetric reinjection, under the influence of a lateral flow, the droplets were deformed and their trajectory deviated from the central position. This led to a short-term change in the Taylor vortices inside the droplets which allowed their contents to mix, which is what triggers the bioluminescent reaction.

It was shown for the first time that bioluminescent reaction with aldehyde substrate could be performed in droplet reactors generated in a microfluidic chip, and bioluminescent light intensity could be measured for each droplet individually.

The presented microfluidic droplet enzymatic assay can be the component of a multi-sensor bioassay [58]. Compared to some disposable paper-based microfluidic devices for water quality assessment [59,60], droplet-based solutions can give rapid feedback in the continuous mode of water pollution monitoring. The obtained results are applicable for the development of early warning systems at the wastewater discharge sites of industrial enterprises or at water treatment plants in addition to the existing certified methods for the registration of water-soluble pollutants.

A recent review of the latest advances, important design strategies, applications and future prospects of microfluidic chips [61] allows us to be confident that the demonstrated concept will also fit into the considered areas of practical applications in biological and medical metrology.

The developed microfluidic immobilized enzymes reactor in droplets (MIERID) could become the basis for studying biological measures [62], the evolution of ensembles of chemoenzymatic systems in MIERID and allow their joint immobilization with chemical molecules, subcellular structures, individual cells and microorganisms. For biomedical application, cells could be added to the droplets. MIERID in the flow could be further controlled and transformed using digital microfluidic platform applications and separately measured by sensors of various designs.

## Figures and Tables

**Figure 1 micromachines-13-01146-f001:**
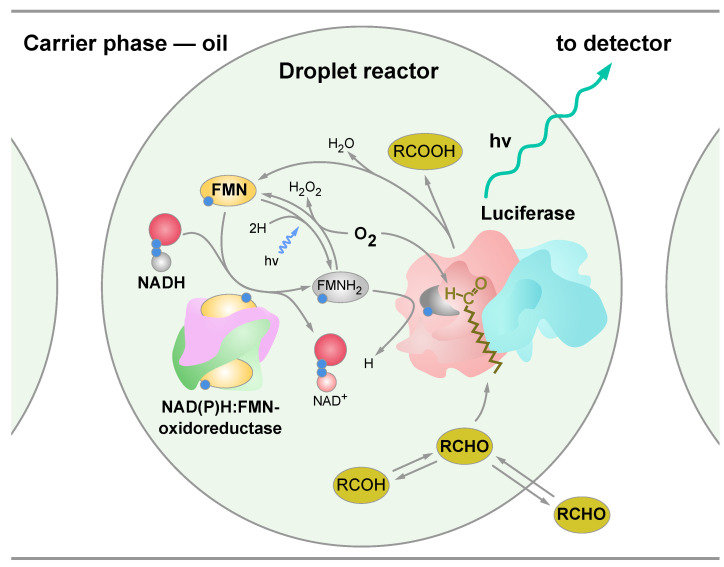
The schematic representation of a droplet reactor with the encapsulated chemoenzymatic system. The multienzyme part of the system consisted of luciferase and NAD(P)H:FMN–oxidoreductase, as well as substrates: flavin mononucleotide (FMN), reduced nicotinamide adenine dinucleotide (NADH) and oxygen.

**Figure 2 micromachines-13-01146-f002:**
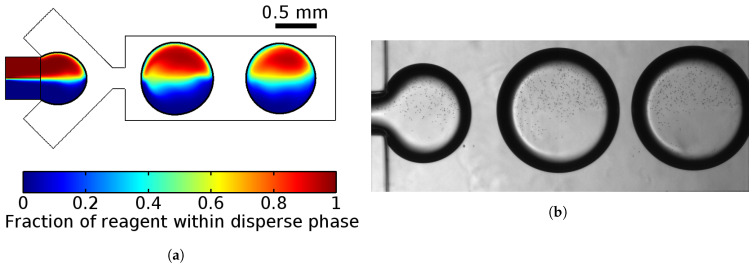
Droplet formation from the two co-laminar flows of the basic and starting solutions. The results of the 2D numerical simulation of the droplet generation with a diameter of 0.8 mm in a channel 1 mm wide with a frequency of 12 drops per minute (**a**) showed that the mixing of laminar flows that are equal in volume was insignificant during the formation of droplets and their further flow along the straight channel. This result was experimentally confirmed for a droplet generator with a channel depth of 0.25 mm (**b**). For visualization, polymer microparticles with a nominal diameter of 1 μm were added to one of the solutions.

**Figure 3 micromachines-13-01146-f003:**
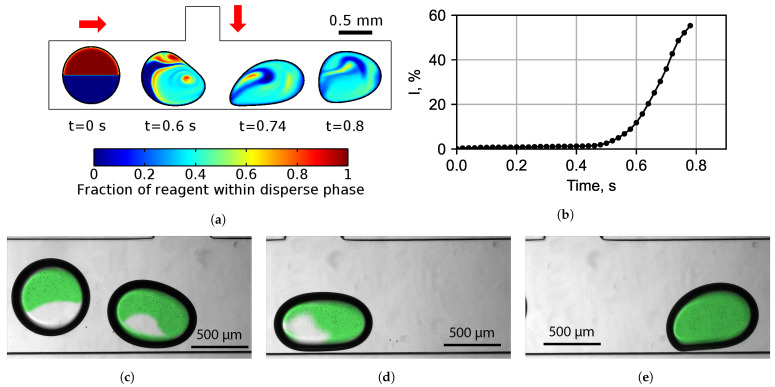
Reinjection of the droplet reactors in an asymmetric T-junction: (**a**) numerical simulations of the droplet flow near the side channel; (**b**) time dependence of the mixing index Equation (Equation 2); and (**c**–**e**) experimental verification in the chip. It can be seen that the polymer microparticles with a nominal diameter of 1 μm in a droplet before the side channel were only located in the zone of the upper Taylor vortex, and after the reinjection, they were distributed over the entire volume of the droplet (flow direction from left to right). For better visualization, the area where the distance between the particles in the figure was less than 100 μm was tinted with green using the ImageJ software.

**Figure 4 micromachines-13-01146-f004:**
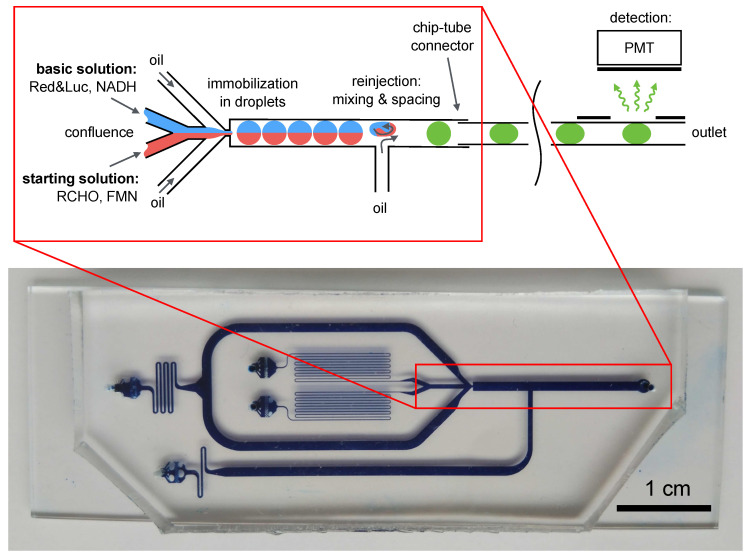
The principle of operation of the microfluidic device for the regular immobilization of Red&Luc with substrates, products and other chemical compounds in droplet reactors (**above**) and the design of the PDMS microfluidic chip (**below**). The channels of the chip presented in the figure were filled with Coomassie Brilliant Blue G-250 for better visualization.

**Figure 5 micromachines-13-01146-f005:**
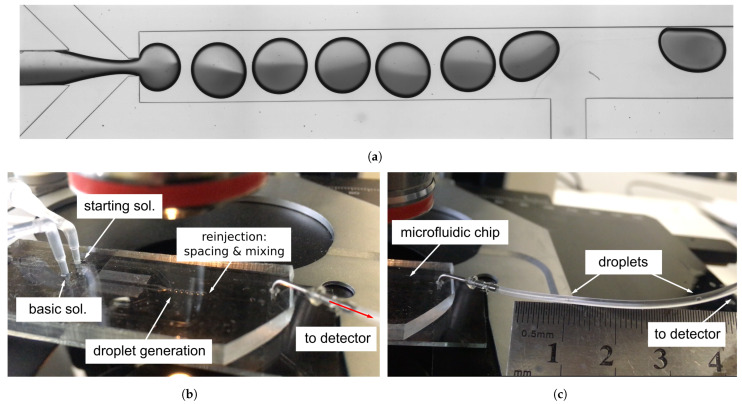
Assembled, tuned and functioning microfluidic device for the regular immobilization of the multienzyme system in droplet reactors (also see Appendix A). The developed microfluidic chip (**a**) consisting of Y- (behind the frame of the picture on the left), X- and T-crosshairs, in which the basic and starting solutions are coupled, droplets are formed, mixed and distanced. One of the components of the dispersed phase was colored with Coomassie Brilliant Blue G-250 for better visualization. The fluid supply pressures were selected to ensure the generation of droplets in a volume of 0.13 μL with a frequency of 12 drops per minute (**b**), and allowing the droplets to move away from each other at a distance of 2–2.5 cm (**c**).

**Figure 6 micromachines-13-01146-f006:**
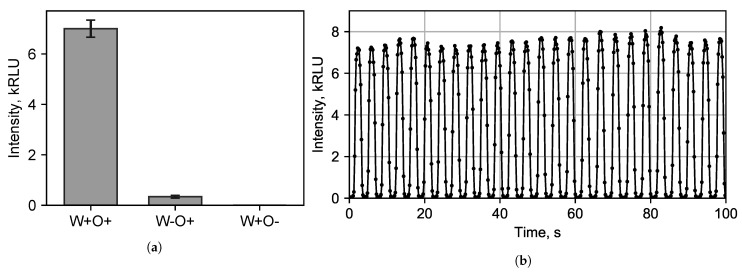
(**a**) Dependence of luminescence intensity on the presence of the tetrodecanal (+/−) in the carrier (O) and dispersed (W) phases. In the presence of tetradecanal (+), its concentration in the carrier and dispersed phase was 10 mM and 0.33 mM, respectively. (**b**) The intensity of luminescence in time was recorded by the photomultiplier tube (PMT) 20 seconds after the start of the enzymatic cascade. Flares with a glow intensity in the region of 6–8 kRLU were recorded at the moments when the droplet was opposite the PMT aperture. When the droplet was carried away with the flow and only the carrier phase appeared at the aperture, the glow intensity corresponding to the level of the PMT dark noise was recorded (approximately 20 RLU).

**Figure 7 micromachines-13-01146-f007:**
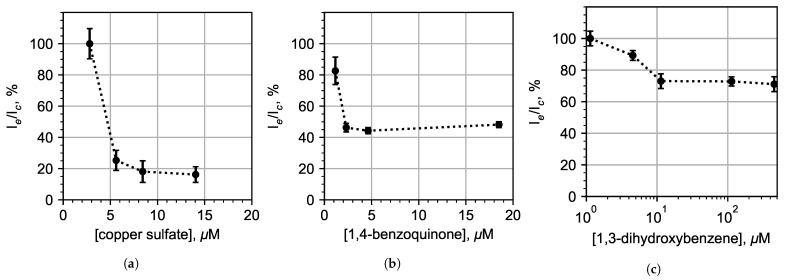
Luminescence intensity inhibition of the multienzyme bioluminescent system in droplets with the addition of model pollutants: (**a**) copper sulfate; (**b**) 1,4-benzoquinone; and (**c**) 1,3-dihydroxybenzene.

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
