# Peer review of "Droplet Microfluidic Device for Chemoenzymatic Sensing"

_micromachines, 2022, doi:10.3390/mi13071146_

Round 1

Reviewer 1 Report

Please justify, why you decided to use/focus on NAD(P)H:FMN-oxidoreductase- 59 luciferase bioluminescent reaction and what alternatives exist!

2.2. Preparation of basic and starting solutions - where does the "recipe" comes from, own delevopment or instruction from the reagents?

Justify why numerical simulations where necessary, which is not obvious.

Fig. 4: add a scale! too much text in teh caption - put in in the main text.

Reviewer 2 Report

The manuscript by Yakimov et al. describes a new microfluidic device that uses emulsion droplets for the chemoenzymatic biosensing of certain pollutants in water. Although the technique described and results produced are very interesting to the readers of Micromachines, the report as it currently stands is not of sufficient quality to warrant publication due to the reasons mentioned below:

The manuscript should be revised improve the quality of English being used for a scientific paper. There are some spelling and grammatical mistakes that should be avoided (e.g. “continues” line 38, “raising” line 41, “hight-throughtput” line 50, “t asymmetry” Figure 3, “we it” line 257, etc.)

In the abstract, the limit of detection should be mentioned for copper sulfate as 5 mg/L and not just 1 mg/L for copper ions; otherwise the authors are required to take into account the molecular weight and show the calculation for the limit of detection of the copper ions in the text. The authors are advised to be consistent throughout the text as figure 7 also uses the concentration of copper sulfate used and not that of copper ions.

Also, the authors have to be consistent with the names of pollutants used in this study. In lines 66-67, the authors mention that “model pollutants (copper sulphate, resorcinol and benzoquinone)”. However, in lines 283-284, the authors mention “model pollutants copper sulfate (heavy metal salt), hydroquinone (phenol), and benzoquinone (quinone) were used.”

In section 2.2 “Preparation of basic and starting solutions”, the authors used very specific figures in the preparation of their solutions i.e. “0.018 activity units” or “888 μl” and “63 μl” for their volume solutions. Is there a reason for not going into nominal rounded numbers such “0.02 activity units” or “890 and 65 μl” to simplify the preparation procedure? Or were these values previously optimized? Authors are required to clarify in text why they used these values; otherwise, they can refer to pervious work if they were previously optimized. 

In lines 104-105, the authors claim that “the inner surfaces of microchannels were coated with a hydrophobic agent.” The authors are required to mention what agent was used and how it was applied to allow the independent replication of their work. The authors are also required to comment on how stable the agent is and if it breaks down or wash away with continuous or repetitive use. Also, does this agent interfere with the detection results of the pollutants in the study?

Authors need to be consistent in their use of decimal separators. In section 2.2, they used decimal points whereas in section 2.4, they use decimal commas. The same goes for consistently using either uL or nL for the droplet volume throughout the text.

Authors are advised to be consistent with the system of units being used in the paper. The value 1/32 can be added between brackets after using the SI equivalent in line 130 “Tubing capillaries with an inner diameter of 1/32 inch”, lines 130, 137 and elsewhere to be consistent with the SI system of units used in the paper.

Authors are required to further elaborate on their statement in lines 133-134 “Therefore, the proportions of solutions were controlled by the difference in their supply pressures.” How did authors check that the ratios were consistently maintained in all droplets? What was the variation and what would be considered an unacceptable level of deviation? The authors can consider adding a figure to show this in section 3.1.1.  

In line 161, the authors claim that the reaction “decays to emit light with a maximum intensity at 490 nm.” What were the wavelengths of the light signal that the photosensitive detector collects and that was used in the analysis? Also, as a suggestion, authors can consider analyzing their light signal at the wavelength of the maximum intensity of reaction as this was previously shown to enhance the limits of detection and quantification of the analyte of interest:

Charbaji et al. (2021) A New Fibrous Material with Embedded Zinc Particles, Eng. Sci. Technol. an Int. J. https://doi.org/10.1016/j.jestch.2020.09.005

Charbaji et al. (2021) A New Paper-Based Microfluidic Device for Improved Detection of Nitrate in Water, Sensors. https://doi.org/10.3390/s21010102

In section 3.1.1, authors are encouraged to add a figure to visually show how strong the linear dependence of droplet diameter on the ratio of dispersed and continuous phases is. 

In section 3.1.2, authors are required to clarify and elaborate on their statement: “a mixing index of up to 60% and a high mixing rate of 0.2% per millisecond even after passing through the side channel, when the liquid circulation in the droplet again became symmetric.”

In lines 221-223, the authors claim that simulation predicts that “By adjusting the pressure in the side channel of the reinjector, the distance between the droplets could be varied in the range of 0.1–5 mm.” However, in line 243 and Figure 5, they claim that fluid pressure was selected to “allowing the droplets to move away from each other at a distance of 2–2.5 cm”. It makes it seem like there’s a discrepancy between simulation prediction and physical results. Authors are required to clarify this point. Why wasn’t the “reduced pressure source” that is mentioned in line 230 used in the simulations. How does its use impact simulation accuracy or results?  

In line 265-266, the authors mention that “The concentration of tetradecanal in the carrier phase was 10 mM, in the dispersed phase it was 0.33 mM”. Whereas, the caption of Figure 6 mentions “In the presence of tetradecanal (+), its concentration in the carrier phase was 10 mM, in the dispersed — 0.6 μM.” The authors are encouraged to be consistent and to make this clear or add the dilution factor to figure 6 to reduce the possibility of misinterpretation.

In figure 6, the authors claim that “Flares with a glow intensity in the region of 6–8 kRLU were recorded at the moments when the droplet was opposite the PMT aperture.” And that the “PMT dark noise was recorded (about 20 RLU).” Then what do the several points below the 6 kRLU mark correspond to? Authors are required to clarify this. If the signal varies from just below 1 kRLU to 8 kRLU as the droplet passes through the PMT channel, how can confident analyte detection take place?

In Figure 7, authors use uM as the unit of pollutant concentration; however, in section 3.3, they use mg/L in their analysis of the results of figure 7. Authors have to be consistent with their use of units throughout the text. 

Why was distilled water used in the pollutant detection experiments instead of ASTM type 1? Using a standardized water sample allows comparison of independent results to their work.

Authors are required to clarify the LOD calculation method used in section 3.3. The limit of quantification (LOQ) should be included. The authors can use plus (or minus) 10 times the standard deviation to the mean of the blank to calculate the LOQ. There are also several other ways and the authors are referred to the below article by Belter et al. for a list of different methods that can be used to calculate and report on the LOQ. 

Belter et al. (2014) Over a century of detection and quantification capabilities in analytical chemistry—Historical overview and trends, Talanta. https://doi.org/10.1016/j.talanta.2014.05.018

Finally, I recommend the authors to make the article easier to follow by the reader by being consistent in their use of compound names or units throughout the text. I also encourage them to include more details and information that would make it easier for the reader to fully understand the approach and steps followed to “independently” reproduce the results.

Reviewer 3 Report

This is a very well written paper. I enjoyed reading it. Some minor points:

Line 35: This -> These

line 40: However -> However,

What I would have appriciated, if more details would have been given on the dependencies on the reinjection rate. The paper is really thin on this point. If a graph showing mixing length in dependence on reinjection flow rate could be added, it will improve the paper quite some bit.

Round 2

Reviewer 2 Report

The authors have thoroughly improved the manuscript according to initial comments and recommendations provided earlier. The authors are encouraged to address the following very minor recommendations:

Few typos and grammatical mistakes are still present (e.g. “ratio vs. ration” line 200)

In line 29, it seems a word or phrase is missing before the word “enzymatic” in “biochemical reaction after enzymatic probes.”

The authors statement in lines 186-187 “Therefore, improvement of reagents mixing might be required.” is unclear. Is this required for future work or has this been already accomplished using the T-junction/T-cross in this work? Authors are encouraged to rewrite this sentence to make clearer what they actually mean. 

In line 202, the authors are encouraged to add a reference to their prior work (Sci Rep. 2021. Vol. 11, â„– 1. P. 8797) after their statement on a similar observation on linear dependence by simulation.  
